# Implementation of medicines pricing policies in sub-Saharan Africa: protocol for a systematic review

Tolib Mirzoev [1], Augustina Koduah,[2] Anna Cronin de Chavez,[1] Leonard Baatiema,[3] Anthony Danso-Appiah,[3] Tim Ensor [1], Irene Akua Agyepong,[4] Judy M Wright [5], Irene A Kretchy,[2] Natalie King[5]

TM and AK are joint first authors.

¹Nuffield Centre for International Health and Development, University of Leeds, Leeds, UK
²School of Pharmacy, University of Ghana, Legon, Greater Accra, Ghana
³School of Public Health, University of Ghana, Accra-Legon, Ghana
⁴Research and Development Division, Ghana Health Service, Accra, Greater Accra, Ghana
⁵Leeds Institute of Health Sciences, University of Leeds, Leeds, UK

**Correspondence to**
Professor Tolib Mirzoev;
t.mirzoev@leeds.ac.uk

## ABSTRACT

**Introduction** Ensuring universal availability and accessibility of medicines and supplies is critical for national health systems to equitably address population health needs. In sub-Saharan Africa (SSA), this is a recognised priority with multiple medicines pricing policies enacted. However, medicine prices have remained high, continue to rise and constrain their accessibility. In this systematic review, we aim to identify and analyse experiences of implementation of medicines pricing policies in SSA. Our ambition is for this evidence to contribute to improved implementation of medicines pricing policies in SSA.

**Methods and analysis** We will search: Medline, Web of Science, Scopus, Global Health, Embase, Cairn.Info International Edition, Erudit and African Index Medicus, the grey literature and reference from related publications. The searches will be limited to literature published from the year 2000 onwards that is, since the start of the Millennium Development Goals.
Published peer-reviewed studies of implementation of medicines pricing policies in SSA will be eligible for inclusion. Broader policy analyses and documented experiences of implementation of other health policies will be excluded. The team will collaboratively screen titles and abstracts, then two reviewers will independently screen full texts, extract data and assess quality of the included studies. Disagreements will be resolved by discussion or a third reviewer. Data will be extracted on approaches used for policy implementation, actors involved, evidence used in decision making and key contextual influences on policy implementation. A narrative approach will be used to synthesise the data. Reporting will be informed by the Preferred Reporting Items for Systematic Reviews and Meta-Analyses Protocols guideline.

**Ethics and dissemination** No ethics approvals are required for systematic reviews.
Results will be disseminated through academic publications, policy briefs and presentations to national policymakers in Ghana and mode widely across countries in SSA.

**PROSPERO registration number** CRD42020178166.

## INTRODUCTION
The current agenda of Universal Health Coverage highlights the importance of access

### Strengths and limitations of this study

► This systematic review protocol follows the Preferred Reporting Items for Systematic Reviews and Meta-Analyses Protocols guidelines.
► The review addresses a gap in the current knowledge of the determinants and outcomes of successful implementation of medicines pricing policies in sub-Saharan Africa (SSA).
► The focus on SSA will help with transferability of lessons across the different countries within the region, though this may lead to omission of important experiences for example from Asia and Latin America and may limit transferability of lessons outside the SSA.
► The search will be restricted to peer-reviewed published articles and grey literature, thus, relevant theses and conference abstracts are likely to be omitted and may affect the depth of evidence on the topic.
► The narrative synthesis approach reflects the nature of published evidence on the topic of policy implementation with no meta-analysis possible, and is a potential limitation of this review.

to safe, quality and affordable medicines as its key driver.[1–3] Increasing access to essential medicines through medicines pricing interventions is an issue of current health policy discourses.[4 5] In response, various policy initiatives have evolved to regulate medicine pricing and improve access.

Globally, different medicine pricing models and strategies exist. These include: generic or biosimilar price linking to originator products, non-proprietary prescribing and generic substitutions, tendering and pooled procurements, internal reference pricing, external price referencing or international price comparisons and managed-entry agreements.[6–8] Implementation of these medicine pricing policies may be dependent on in-country manufacturing capacity, pricing levels of the medicines, whether medicines



are generic or branded, and whether medicines are for outpatient or inpatient services.[9]

Many of these medicine pricing policies are being implemented in high-income countries. However, unlike in high-income countries, low-income and middle-income countries have less regulated and developed pharmaceutical markets and have different challenges in distribution and production.[1] In light of this, multiple medicine pricing models and strategies are required to achieve equitable access to safe, quality and affordable medicines,[10] particularly in sub-Saharan African (SSA) countries.[10]

## Rationale

Ensuring availability and accessibility of medicines is an important mechanism by which national health systems can equitably address health needs of their populations, including the poorest and the most vulnerable. In SSA, this is a recognised policy priority. For example, in the last two decades different medicines pricing policies were implemented in South Africa[11][12] and between 2012 and 2017, the Government of Ghana introduced four policies to improve access to medicines through medicine price regulation, and ultimately, health outcomes and quality of life. These policies are currently at different stages of their implementation and despite these efforts, medicine prices have remained high and continue to rise, making them inaccessible to a large proportion of populations. This raises questions as to why and how these policies are failing to achieve the desired outcomes.

In this systematic review, we will explore the effectiveness of implementation of medicines pricing policies in the SSA context. We want to identify which policies have been implemented and then explore three broad dimensions of their implementation. First, we want to understand what happened, that is, identify evidence on effective implementations of medicines pricing policies reflected in a reduction in prices and improvement in access to medicines and subsequently healthcare. Second, we want to understand how it happened, that is, we want to identify and unpack the implementation processes and approaches deployed in terms of their timing, participation of actors and role of evidence. Third, we want to understand why it happened. We want to identify and synthesise key reported facilitators and barriers to the implementation and understand how they affected the implementation of these policies within their respective contexts.

## Aim and objectives

In this systematic review, we will address the following overall question: what are the key determinants of implementation of medicines pricing policies in SSA countries? More specifically, we will answer four questions:

1. Which medicines pricing policies have been implemented in SSA and what are their key elements?

2. How have these policies been implemented (in relation to implementation approaches, processes, involvement of actors, role of evidence, etc)?
3. Which key facilitators and barriers affected the implementation of medicines pricing policies, and how?
4. Which implementation of medicines pricing policies in SSA are effective (in relation to reducing prices of medicines and improving access to services)?

This review is being undertaken during April 2020–May 2021 as part of the project on 'Improving equitable access to essential medicines in Ghana through bridging the gaps in implementing medicines pricing policy' (AMIPS project)—an National Institute for Health Research (NIHR) funded award received jointly by the University of Leeds, University of Ghana and the Ghana Health Service. The results of this review will be combined with results of policy analyses in Ghana and will inform engagements with key stakeholders on improving the implementation of the current policies and identification of future research and development priorities. Our ambition is for evidence from this review to contribute to improved implementation of medicines pricing policies across countries of SSA.

This protocol follows the Preferred Reporting Items for Systematic Reviews and Meta-Analyses-Protocols (PRISMA-P) guidelines[13] and a PRISMA-P checklist is available as an online supplemental file.

## METHODS AND ANALYSIS

### Eligibility criteria

#### Studies

We will include empirical studies including Randomised Controlled Trials, quasi-experimental studies and cohort and cross-sectional studies. Reviews (scoping reviews, meta-syntheses, realist syntheses) will also be included and individual primary studies from the systematic reviews will be manually included as empirical literature. We will exclude opinion pieces and conceptual/theoretical publications which do not report documented empirical data from either primary studies or reviews.

Specific inclusion criteria will be: (1) focus on the medicines pricing policies that is, policies, strategies, interventions or plans which aim to improve affordability of medicines in the country. The link to improvements in access to healthcare may be implicit and is not a requirement; (2) focus on policy implementation, that is, either as part of the whole policy process (agenda-setting, development, implementation) or as an exclusive focus; (3) SSA country contexts, that is, either as part of the comparative studies or as a sole focus; (4) studies which were published since the agenda of Millennium Development Goals (MDGs) was initiated shortly before 2000 and (5) papers with relevant information available for analysis.

Specific exclusion criteria are: (1) policy analyses which focus solely on policy agenda-setting and development stages of the policy process; (2) studies from high-income country contexts and outside SSA; (3) studies conducted

2 years or more prior to 2000 but published after 2000 will be excluded in consideration of MDGs and Sustainable Development Goals (SDGs) agenda which started in 2000; (4) papers in languages where we are unable to have the resources for translation (the team has access to French, Spanish and Russian-speaking researchers) and (5) papers with no full text available for analysis.

### Participants
The participants to be covered in this review will be: policy-makers, implementers, service providers, patients and beneficiaries of successful implementation of medicines pricing policies (of any gender, age, ethnicity, socioeconomic group, health status or urban–rural residence).

### Interventions
Implementation of medicines pricing policies, that is, policies, strategies, interventions or plans which aim to improve affordability of medicines in the country.

### Comparison
No comparison or control is applicable to this study.

### Outcomes
Successful implementation will be measured as reduction in medicines prices, and improved access to medicines along the supply chain. Any studies describing unsuccessful implementation will also be used to inform the lessons learnt.

### Study records
#### Searches
We will search the following databases: Medline (1946–present), Web of Science (1990–present), Scopus (1823–present), Global Health (1973–present), Embase (1947–present), Cairn.Info International Edition (all available years), Erudit (all available years) and African Index Medicus (all available years). The Medline search strategy is available as an online supplemental file. The search strategies will incorporate index terms from Medical Subject Headings (MeSH) and text words for the search concepts:
1. Sub-Saharan African Countries. This will include terms/synonyms for sub-Saharan Africa AND list of individual countries in the region.
2. Drug/Medicines pricing. This will include terms/synonyms for: medicines / pharmaceuticals / drugs / prescriptions AND pricing / cost / affordability / fees / purchase / rebate(s) / tariffs / incentives / benchmarking / reference pricing / payment / spend* / expenditure / subsid* / procurement.
3. Policy. This will include terms/synonyms for: policy / strategy / plan / framework / regulations / guidelines / rules / intervention / tax / exemption.
4. Implementation. This will include terms / synonyms for: Implementing / Implement(s) / Implementation approach(es) / Process(es) / Facilitator(s) / Barrier(s) / Factors / Determinants / context.

The searches will be limited to the literature published from the year 2000 and onwards. This is in consideration of the Millennium Development Goals agenda which started in 2000 with a clear focus on improving access to medicines and services. We will follow up on the references to the individual studies as required. We will manually search for the included references in relevant retrieved reviews (systematic reviews, scoping reviews, meta-syntheses, realist syntheses) for additional relevant studies for inclusion. In addition, we will search grey literature including global development websites: World Health Organisation's (WHO) Institutional Repository for Information Sharing (IRIS), World Bank, Knowledge, Evidence and Learning for Development (K4D) repository, Gates Foundation and contacts with experts in the field.

### Data management
We will upload all references identified through searches (electronic database and additional searches) into Endnote version X9. Once duplicates are removed, the remaining references will be exported into Rayyan (https://rayyan.qcri.org/welcome), an online free systematic review tool for screening.

### Screening
Titles and abstracts will be divided up across the review team and screened individually for eligibility using prespecified eligibility criteria flow chart, which is available in an online supplemental file. At least 20% of individually reviewed titles and abstracts will then be cross-checked by at least two members of the team. Full texts will be obtained for all the potentially relevant studies and screened by two members of the team independently, and disagreements will be resolved through discussion. Where necessary, a third member of the team will engage to help resolve disagreements.

### Data extraction
The following data will be extracted by two members of the review team into an appropriate data extraction form:
► Article information (full citation, year study was conducted, study type, setting / country).
► Medicine pricing policies studied (including which key elements the policies included).
► Documented effects on prices of medicines (including how identified and reported).
► Effects on access to medicines (including how identified and reported).
► Effects on access to healthcare (including how identified and reported).
► Implementation approach (including processes, actors involved and their roles and evidence used to inform implementation).
► Key influences on policy implementation (including facilitators and constraints and how they affected implementation).

## Quality assessment and risk of bias

Quality of each included study will be appraised. We will use validated quality assessment tools and the critical appraisal tools for relevant studies (qualitative and quantitative research) from the Joanna Briggs Institute https://joannabriggs.org/ebp/critical_appraisal_tools. While at this point, we do not intend to change the actual criteria, the interpretation and application of the tools will be within the context of our study which focuses on key determinants of effective implementation of medicines pricing policies in SSA context. For example, clarity of focus will be assessed in relation to how the different aspects of policy implementation (processes, use of evidence, involvement of actors) are identified and consistently used in the reviewed papers.

A careful assessment of risk of bias in the included studies will be performed by two reviewers, who will first independently assess the quality of each study against each criterion. Results will be shared and agreed, and any disagreements will be addressed through engaging a third reviewer.

## Data synthesis and interpretation
### Strategy for data synthesis

The main outcome in our study is the medicine pricing policy implementation. Policy implementation is typically done within a single country, but where the same policy is implemented in different countries, the analysis will take the specific context of the country into consideration.

In exploring the policy implementation, we will employ established policy theories and frameworks such as Walt and Gilson's policy triangle,[14] Baumgartner and Jones's punctuated equilibrium[15] and Lipsky's street-level bureaucracy,[16] and will also draw on further theories and frameworks developed or adapted within the reviewed papers.

Where possible, we will compare the effects of the policies in a quantitative synthesis. We anticipate, however, that the heterogeneity of reporting of outcomes and of context may make it impossible to conduct a meta-analysis. In such a situation we will focus on narrative synthesis.

While using qualitative or narrative synthesis approach,[17] data related to the medicines pricing policies will be extracted from the Introduction, Methods and possibly Results sections. Data on the effects of policy implementation and the policy implementation approaches, and key influences will be extracted from the Results and Discussion sections. Extracted data will be analysed thematically and will be structured around the specific questions of the review.

The interpretation of the results will follow the identified themes for each review question. For example, in answering the third review question we will divide the factors into facilitators and constraints and potentially will further subdivide them by their nature (eg, community issues, health systems issues, wider socioeconomic influences).

At the moment, we are not planning analysis of subgroups or subsets. However, depending on the breadth of extracted data we may consider subgroups such as geographical region (West Africa, East Africa, Southern Africa), setting (urban, rural) or categories of implementers (health facilities, pharmacies).

The cumulative strength of body of evidence will be assessed across the risk of bias and consistency, drawing on relevant approaches such as Grading of Recommendations Assessment, Development and Evaluation (GRADE).

## Patient and public involvement
No patient involved.

## ETHICS AND DISSEMINATION
Ethics approvals are not required for systematic reviews. However, ethics approvals for the wider AMIPS study within which this review is being undertaken have been granted by the ethics committees from the Ghana Health Service (ref GHS-ERC006/02/20) and the University of Leeds School of Medicine (ref MREC 19–060).

We will disseminate results through academic papers and stakeholder workshops in Ghana and other SSA countries where possible. In Ghana, the review results will be complemented by reviews of policy documents. The findings of this review will also be presented at scientific conferences such as the biannual Global Symposia on Health Systems Research and Thematic Working Groups of the Health Systems Global.

The results of this review will inform empirical investigations of implementation of medicines pricing policies in Ghana through the in-depth interviews and focus groups, and engagements and consultations with policymakers on seeking ways of further improving the implementation of medicines pricing policies in Ghana.

**Contributors** AK and TM jointly conceived the study; TM, AK, ACdC, LB, AD-A, TE, IAA, JMW, IAK and NK contributed to the review design and jointly wrote the protocol; TM, AK, ACdC, LB, AD-A, TE, IA, JMW, IAK and NK read and approved the final version of the manuscript. AK and TM will be the guarantors of the review.

**Funding** This research was commissioned by the National Institute for Health Research (NIHR) NIHR Global Health Policy and Systems Research Development Award using UK aid from the UK Government (grant number 130219). The views expressed in this publication are those of the author(s) and not necessarily those of the NIHR or the Department of Health and Social Care.

**Competing interests** None declared.

**Patient consent for publication** Not required.

**Provenance and peer review** Not commissioned; externally peer reviewed.

**ORCID iDs**
Tolib Mirzoev http://orcid.org/0000-0003-2959-9187
Tim Ensor http://orcid.org/0000-0003-0279-9576
Judy M Wright http://orcid.org/0000-0002-5239-0173

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
