## [Reviewer comments · BMJ Open]

ARTICLE DETAILS

TITLE (PROVISIONAL)	Implementation of medicines pricing policies in sub-Saharan Africa: protocol for a systematic review
AUTHORS	Mirzoev, Tolib; Koduah, Augustina; Cronin de Chavez, Anna; Baatiema, Leonard; Danso-Appiah, Anthony; Ensor, Tim; Agyepong, Irene; Wright, Judy; Kretchy, Irene; King, Natalie

VERSION 1 – REVIEW

REVIEWER	Varsha Bangalee University of KwaZulu-Natal South Africa
REVIEW RETURNED	02-Oct-2020

GENERAL COMMENTS	The reviewer is of the opinion that the anticipated systematic review will be of great value and should definitely be pursued. The only minor comments are that there are several areas where the grammar in the submitted article particularly in the introduction section can be improved. The other small concern is that the authors keep stating that they will report on the successes of policy implementation, but in the review they are not planning to only report successful policies, but all pricing policies, so perhaps there needs to be a quick review of the aims, objectives and research questions to ensure that they are clear and better aligned.
---

REVIEWER	Dr. Felix Khuluza University of Malawi-College of Medicine, Malawi
REVIEW RETURNED	19-Oct-2020

GENERAL COMMENTS	There is need to relook at the title. The title is not specific enough. The systematic is meant to inform pricing policy guidance to Ghanaian authorities, and this has to be captured in the title.
--

VERSION 1 – AUTHOR RESPONSE

Reviewer 1 (Varsha Bangalee)	
The reviewer is of the opinion that the anticipated systematic review will be of great value and should definitely be pursued. The only minor comments are that there are several areas where the grammar in the submitted	Thanks you and we also feel this review is important. We have now carefully proof-read the manuscript and corrected grammar throughout.

article particularly in the introduction section can be improved.	
The other small concern is that the authors keep stating that they will report on the successes of policy implementation, but in the review they are not planning to only report successful policies, but all pricing policies, so perhaps there needs to be a quick review of the aims, objectives and research questions to ensure that they are clear and better aligned.	We take this point and have now removed successful from the overall review question (p.5) so it is now more neutral and consistent with review questions. You are correct, we will not focus only on cases of successful implementation (even though in reality most published papers are).
Reviewer 2 (Dr. Felix Khuluza)	
There is need to relook at the title. The title is not specific enough. The systematic is meant to inform pricing policy guidance to Ghanaian authorities, and this has to be captured in the title.	This protocol covers the systematic review only and not the wider AMIPS study. Our review questions (p5) do not include informing policy guidance in Ghana. Therefore, we'd like to keep the current title. However, in addressing this comment we have modified the Introduction section of the Abstract (p.2) and the explanatory paragraph after the Aim and Objectives (p.5) to explain that it is our ambition for evidence from this review to contribute to improved policy implementation in countries of SSA.